# Control of structural flexibility of layered-pillared metal-organic frameworks anchored at surfaces

Suttipong Wannapaiboon[1,2], Andreas Schneemann [1], Inke Hante[3], Min Tu[3,4], Konstantin Epp[1], Anna Lisa Semrau [1], Christian Sternemann[5], Michael Paulus[5], Samuel J. Baxter[6], Gregor Kieslich[1] & Roland A. Fischer [1]

Flexible metal-organic frameworks (MOFs) are structurally flexible, porous, crystalline solids that show a structural transition in response to a stimulus. If MOF-based solid-state and microelectronic devices are to be capable of leveraging such structural flexibility, then the integration of MOF thin films into a device configuration is crucial. Here we report the targeted and precise anchoring of Cu-based alkylether-functionalised layered-pillared MOF crystallites onto substrates via stepwise liquid-phase epitaxy. The structural transformation during methanol sorption is monitored by in-situ grazing incidence X-ray diffraction. Interestingly, spatially-controlled anchoring of the flexible MOFs on the surface induces a distinct structural responsiveness which is different from the bulk powder and can be systematically controlled by varying the crystallite characteristics, for instance dimensions and orientation. This fundamental understanding of thin-film flexibility is of paramount importance for the rational design of MOF-based devices utilising the structural flexibility in specific applications such as selective sensors.

[1] Chair of Inorganic and Metal-Organic Chemistry, Department of Chemistry, Technical University of Munich, Lichtenbergstraße 4, 85787 Garching, Germany. [2] Synchrotron Light Research Institute (Public Organization), 111 University Avenue, Muang District, Nakhon Ratchasima 30000, Thailand. [3] Chair of Inorganic Chemistry II, Faculty of Chemistry and Biochemistry, Ruhr-University Bochum, Universitätstraße 150, 44801 Bochum, Germany. [4] Centre for Surface Chemistry and Catalysis, Katholieke Universiteit Leuven, Celestijnenlaan 200fBox 24613001 Leuven, Belgium. [5] Fakultät Physik/DELTA, Technische Universität Dortmund, Maria-Goeppert-Mayer Straße. 2, 44227 Dortmund, Germany. [6] School of Chemistry and Biochemistry, Georgia Institute of Technology, Atlanta, GA 30332, USA. Correspondence and requests for materials should be addressed to R.A.F. (email: roland.fischer@tum.de)

Flexible materials are intriguing for the development of novel composites, due to their responsiveness to specific stimuli by undergoing structural transformation[1–5]. For centuries, the potential utility of biological flexible materials has been perceived. Leonardo da Vinci produced the first hygrometer by employing the expansion/contraction of human hair upon ad- and de-sorption of moisture[6]. More advanced responsiveness has been observed in biomimetic and organic materials such as enzymes, biomolecules and stimuli-responsive polymers. These materials can change their conformation in response to external stimuli and adapt their structures to surrounding environments. Consequently, they can transduce chemical and biological signals into optical, electrical and mechanical signals leading to various applications such as smart optics, biosensors and electro-mechanical systems[5]. Even though flexibility is quite common in elastic polymers, creating flexible, crystalline solid-state materials remains a challenge, since their long-range compositional ordering may prevent spatial rearrangement. Only few examples of zeolites show some structural adaptivity in response to high-pressure compression, however still depending on the hydrostatic pressure-transmitting media[7,8]. However, recently, structural changes in elastic crystals of copper(II) acetylacetonate during mechanical stress have been reported[9].

Metal-organic frameworks (MOFs) are a promising class of porous, crystalline materials, formed by the combination of metal ions or clusters and multitopic organic linkers, which allows to integrate flexibility and responsivity into rigid, crystalline structures[10]. Due to the versatility of precursor building blocks and the possibility for chemical functionalisation of coordination space, MOFs offer a broad perspective for rationally designing and fine tuning their structure, characteristic features and chemical properties of the pore surface[11–14]. The specific design of MOF components to control attractive forces used for assembling the porous scaffolds can initiate structural dynamics in some MOFs (named flexible MOFs or soft porous crystals (SPCs))[10,15–18]. Depending on their characteristic features, SPCs can undergo reversible phase transitions in various flexible modes (i.e., breathing[19], swelling[20], ligand rotation[21], subnetwork displacement[22,23]) upon external stimuli such as guest sorption, temperature change, light and mechanical pressure[15,16]. The discovery of highly-responsive MOFs highlights the potential applications for selective gas storage[24–26], effective gas separation[27], controlled drug release[28] and also smart sensors[29].

The so-called breathing transition in MOFs has been extensively studied for the prototypical class of M(bdc)(OH) frameworks (MIL-53, MIL = Materiaux Institute Lavoisier; $M^{3+}$ = $Al^{3+}$, $Fe^{3+}$, $Cr^{3+}$, $Sc^{3+}$, $Ga^{3+}$, and $In^{3+}$; $bdc^{2-}$ = 1,4-benzenedicarboxylate). These frameworks are built up by one dimensional M–OH–M chains bridged by $bdc^{2-}$ ligands and typically undergo (in the case of M = $Al^{3+}$, $Cr^{3+}$, $Ga^{3+}$) a large pore (lp) to narrow pore (np) phase transition upon guest adsorption in the activated (i.e., solvent free) state[19,30]. However, the utilisation of different metal centres (i.e., $Fe^{3+}$, $Sc^{3+}$)[31,32] and $bdc^{2-}$ derivatives featuring functional groups[33] modifies the breathing effect substantially. These transitions are usually identified and analysed by X-ray diffraction techniques (in-situ and ex-situ) but are also indicated by steps in the adsorption isotherm[16]. Apart from MIL-53, layered-pillared MOFs ($M_2L_2P$; M = metal ion centre; L = linear dicarboxylate linker; P = neutral nitrogen donor pillar) show under particular circumstances breathing transition. The prototypic $Zn_2(bdc)_2(dabco)$ (dabco = 1,4-diazabicyclo[2.2.2]octane) consists of a square-grid $Zn_2(bdc)_2$ lattices featuring $Zn_2$ paddlewheels, bridged by dabco pillars along the axial position of the paddle-wheel. This material reveals small structural transition upon adsorption of certain guest molecules (i.e., DMF and benzene) due to the deformation of the paddle-wheel $Zn_2(bdc)_2$ grids when

interacting with the guests[34], however, if isopropanol is adsorbed inside the structure, a considerable breathing transition is observed[35]. Along these lines, functionalisation of the dicarboxylate linkers can introduce and even tailor the substantial responsiveness[36,37]. We illustrated in our previous works that the incorporation of alkylether-functionalised bdc linkers in $Zn_2(fu-bdc)_2(dabco)$ compounds (fu-bdc = 2,5-alkoxy-functionalised-1,4-benzenedicarboxylate) induces a breathing effect upon polar guest sorption[37] and temperature change[38]. The dynamic breathing phenomenon between the guest-filled lp form and the guest-free np form can be finely tuned by changing the characteristics of the pendant groups[37], and recently also by the choices of metal nodes[39]. Moreover, mixing of different metal nodes or different linkers is another strategy to tune the structural responsiveness[40].

In addition to modifying the MOF composition, controlling their mesoscopic physical form such as crystal size and morphology is known to uniquely affect their properties[41]. Downsizing of flexible MOF crystallites to the nanoscale regime can alter their adsorption profiles and even generate new intrinsic features differing from the bulk MOFs[42–45]. For example, the bulk form of the $Cu_2(bdc)_2(bipy)$ compound (bipy = 4,4′-bipyridine) shows an opened-pore (op) to closed-pore (cp) structural transition upon guest desorption, while the nanocrystalline form (<60 nm) remains in the op after guest removal. Transformation to the cp form is only achieved by thermal treatment[43]. Moreover, DUT-8(Ni) ($Ni_2(ndc)_2(dabco)$; $ndc^{2-}$ = 2,6-naphthalene-dicarboxylate) also exhibits size-dependent framework flexibility upon guest sorption[44]. Small DUT-8(Ni) crystals (<500 nm) retain the op after activation, whereas larger DUT-8(Ni) crystals (>1 μm) undergo the op-to-cp breathing transition after thermal activation[44].

The implementation of MOFs into thin films has received increasing attention over the last few years[46,47], since it is imperative for the advancement of MOF-based devices. Key challenges in their fabrication are the ability to engineer and enhance characteristic MOF features at the nanoscale, and to spatially control crystallisation processes to be compatible with device configurations[46–48]. Among various methods, stepwise liquid-phase epitaxy (LPE) is beneficial for the formation of homogeneous MOF thin films with controllable thickness. Moreover, the crystallite orientation can be controlled within the LPE process by varying the surface functionality (e.g., by using self-assembled monolayers (SAM) of organothiols featuring different terminal-groups)[49]. This method is driven by the alternating immersion of a substrate into solutions containing individual MOF precursor components[49,50]. Even though various MOFs with rigid structures have been fabricated as thin films, studies on flexible MOF thin films remain scarce. The first attempt was the preparation of a $Fe_3O(H_2O)_2X(bdc)_3$ (X = $F^-$ or $OH^-$; MIL-88B (Fe)) film by solvothermal synthesis on a SAM-functionalised Au-coated substrate. This micrometre-sized crystalline film exhibits framework swelling induced by moisture adsorption in a similar manner to the bulk MIL-88B crystal[51]. Kitagawa and co-workers employed the LPE method to fabricate nanometre-thick inter-digitated Hofmann-type MOF thin films, which compellingly exhibit unique structural motifs which influence the adsorption profiles as compared to the bulk MOFs[52,53]. Remarkably, downsizing of the non-porous $Fe(py)_2[Pt(CN)_4]$ (py = pyridine) to a thin film of 16 nanometres thickness initiates a gate-opening structural transformation (cp-to-op) induced by the lower potential energy barrier[53]. At the nanoscale, guests can diffuse in between neighbouring interdigitated layers and move them apart. Increasing the film thickness above a certain threshold leads to a suppression of the flexibility as observed for the bulk material[53].

In depth understanding of the structural responsiveness and other emerging effects when anchoring flexible MOF crystallites

onto substrates is necessary to facilitate the development of flexible MOF-based devices such as highly-selective sensors. Herein, we describe the fabrication of alkylether-functionalised $Cu_2(DE\text{-}bdc)_2(dabco)$ (**1**) and its analogue $Cu_2(BME\text{-}bdc)_2(dabco)$ (**2**) as thin films (DE-bdc = 2,5-diethoxy-1,4-benzenedicarboxylate and BME-bdc = 2,5-bis(2-methoxyethoxy) −1,4-benzenedicarboxylate) on SAM-functionalised Au-coated quartz crystal microbalance (QCM) sensors. The growth behaviours and the adsorption profiles are investigated by tracking the QCM frequency change, which is proportional to the mass change based on the Sauerbrey Equation. A study of guest-induced structural flexibility of the thin-films in comparison with the bulk MOFs is performed by in-situ synchrotron grazing incidence X-ray diffraction (GIXRD) during guest adsorption by a custom-built, semi-quantitative vapour adsorption unit (Supplementary Fig. 4). Notably, unique and different structural responsiveness upon methanol sorption in the thin films is observed. This highlights not only the striking effect of anchoring the flexible MOF crystallites onto the substrate surface, but also the marked dependence on the number of LPE deposition cycles (i.e., film thickness) (Fig. 1).

## Results

**Structural flexibility of bulk powders.** Polycrystalline powder of **1** (**1bulk**) was synthesised and characterised to establish a benchmark data set for comparison of the structural flexibility of **1** in bulk and as a thin film. **1bulk** was prepared by solvothermal reaction of $Cu(NO_3)_2 \cdot 3H_2O$ with $H_2DE\text{-}bdc$ and dabco (Fig. 2a) in DMF at 120 °C for 48 h. Upon activation in vacuo overnight at 130 °C, **1bulk** undergoes a structural transformation from the solvated, **lp** form to the guest-free, **np** form. After solvent removal from the pore of **1bulk**, the neighbouring alkylether-functionalised sidechains can interact with each other. Referring to the literatures[37,39], single crystal XRD was used to identify the structural evidence of the as-synthesised (solvated) form of Zn-based alkoxyether-functionalised layered-pillared MOFs and Pawley refinements were used to identify the change of lattice parameters with respect to the breathing phenomena upon guest adsorption and desorption. This refinement indicated the wine rack-like motion which changes the square $M_2(fu\text{-}bdc)_2$ grids in the solvated, **lp** form into rhombic grids in the activated, **np** form, while the dabco-containing axis remains mostly constant (Supplementary Fig. 5). Herein, we performed Pawley refinements of the lattice parameters of the MeOH-solvated and activated forms of **1bulk** and **2bulk** from their GIXRD patterns (using synchrotron X-ray radiation with a wavelength of 0.827 Å). (Supplementary Figs. 5 and 7, Supplementary Table 1). In good agreement with the literature, we observe from the Pawley refinements that **1bulk** (as well as **2bulk**) shows an increase of lattice parameter $a$ and a decrease of lattice parameter $b$, while lattice parameter $c$ remains constant upon MeOH desolvation. Consequently, these changes of the unit cell parameters lead to a reduction of the total unit cell volume, indicating the guest induced breathing transition (from the solvated, **lp** form to the activated, **np** form)[37,39]. These evidences confirm the structural transition of the copper-based materials **1bulk** and **2bulk** in a similar manner to their zinc analogues, as well as their thin-films during the sorption of methanol.

We further investigated the structural flexibility of **1bulk** as a response to polar-guest sorption (herein, methanol vapour) at ambient temperature by in-situ GIXRD. According to the GIXRD patterns (Fig. 2b), adsorption of methanol at relative vapour pressure $(P/P_0)$ of 10% in a dynamic He stream (blue line plots of GIXRD patterns) initiates the breathing transition in **1bulk** from the activated, **np** form (the main diffraction peak is highlighted

with a yellow bar) to the solvated, **lp** form (highlighted with a green bar). A complete transformation towards the **lp** form occurs after the relative vapour pressure of the methanol stream is increased to 20% $P/P_0$. Purging the MeOH-solvated **1bulk** with dry He gas at 25 °C desorbs methanol and the initial **np** phase is regained (wine-colour line plots of GIXRD patterns). In addition, polycrystalline powder of **2** (**2bulk**) exhibits a similar trend of the structural flexibility upon methanol sorption at 25 °C, even though the change of the refined lattice parameters is different (Supplementary Fig. 10). This observation illustrates a way to modulate the structural responsiveness in MOFs even at room temperature by varying size and functionality of the pendent sidechains, which can interact with the guest molecules[37].

**Fabrication of MOF thin-films and their crystallinity.** Aiming for high-performance MOF-based devices, we herein focus on the implementation of structural flexibility into MOF thin films onto QCM sensors. A stepwise LPE process is used for the fabrication of **1** and **2** thin films with varying deposition cycles, hereafter named as **1tf$_x$** and **2tf$_x$** ($x$ = number of deposition cycles). In the LPE process, solutions of $Cu(OAc)_2$ and the mixed organic linkers are alternatingly dosed to the —COOH terminated SAM-functionalised QCM substrate in a continuous flow reactor for 40, 60, 80 and 120 cycles. Importantly, each precursor dosing step is followed by flushing of the reactor cell with ethanol. The applied LPE procedure is schematically shown in Fig. 3a. The deposition onto a QCM substrate allows for measurements of the change of QCM oscillator frequency and in turn in-situ investigation of the film growth is possible. We observe an effective control of the deposited sample mass on the substrate as a function of deposition cycles, which implies stepwise construction of the MOF crystallites (Fig. 3b). A closer inspection of the QCM frequency change confirms saturation during each deposition step of the metal and the mixed organic linkers, which reflect a self-terminated growth of the layered-pillared MOF thin films (Supplementary Fig. 11).

Considering the LPE growth of **1tf$_x$** on the —COOH terminated QCM substrates, the as-synthesised thin films exhibit a high crystallinity with a preferred orientation along the (110) plane of the solvated, **lp** form (black line plots of the out-of-plane GIXRD patterns illustrated in Fig. 4), referring to the Zn-based analogous structure reported in the literature[37] and the refined lattice parameters of **1bulk** in Supplementary Fig. 6. According to these results, the $Cu_2(DE\text{-}bdc)_2$ grids in **1tf$_x$**, which can exhibit a wine rack-like motion during the breathing phenomenon, are oriented orthogonally to the substrate surface. Moreover, crystallite sizes were calculated using the Scherrer equation from the GIXRD patterns of MeOH-solvated **1tf$_x$** samples (see Supplementary Fig. 12). There is no significant difference of the calculated crystallite size of **1tf$_x$** samples. Note that, Scherrer equation is not applicable for particles with a size larger than about 100 or 200 nm. Considering the particle size within the **1tf$_x$** samples (see SEM images discussed in detail in the following section), the observed particle sizes range within the sub-micrometres scale. Hence, the calculated crystallite sizes of **1tf$_x$** based on the Scherrer equation could be highly deviated from the actual size. Therefore, they cannot be properly used to identify the size difference of the obtained **1tf$_x$** samples and consequently the effect on structural flexibility of the samples reported herein.

**Structural flexibility upon methanol sorption of MOF thin-films.** In order to examine the structural responsiveness of **1tf$_x$** in comparison to **1bulk** and understand the influence of the crystallite dimension on the framework flexibility, GIXRD patterns are

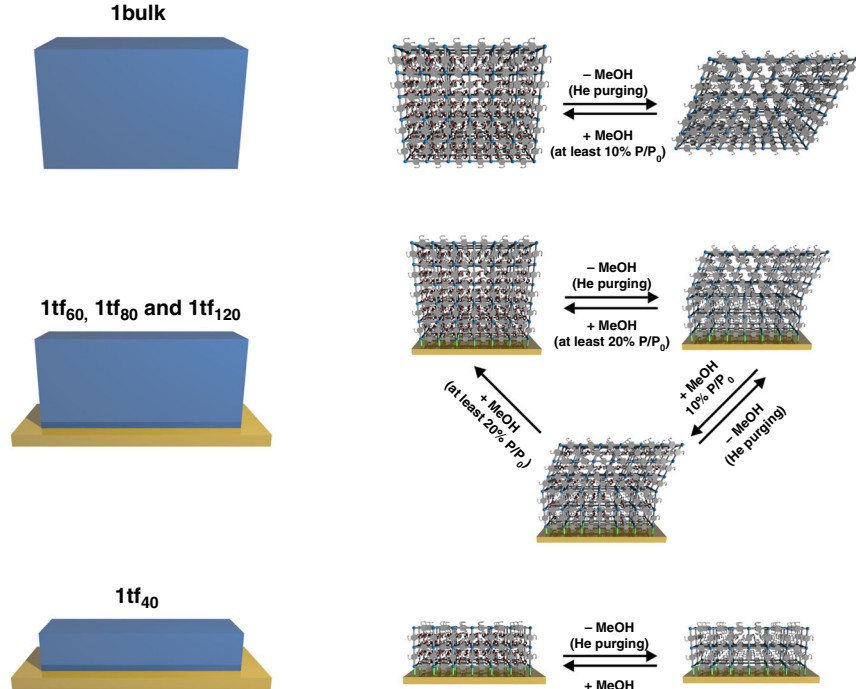

**Fig. 1** Different structural flexibilities upon methanol sorption of $Cu_2(DE\text{-}bdc)_2(dabco)$ (**1bulk**) and thin-films (**1tf$_x$**; x = deposition cycles). Anchoring **1** on the surface affects the structural responsiveness induced by guest sorption, which shows unique breathing behaviour in dependence of the total number of LPE deposition cycles (in other words, the crystallite dimension). Note that, the schematic structures are only suggested structures based on unit cell refinements (Pawley method) of GIXRD patterns of the materials and not a depiction of the actual structures

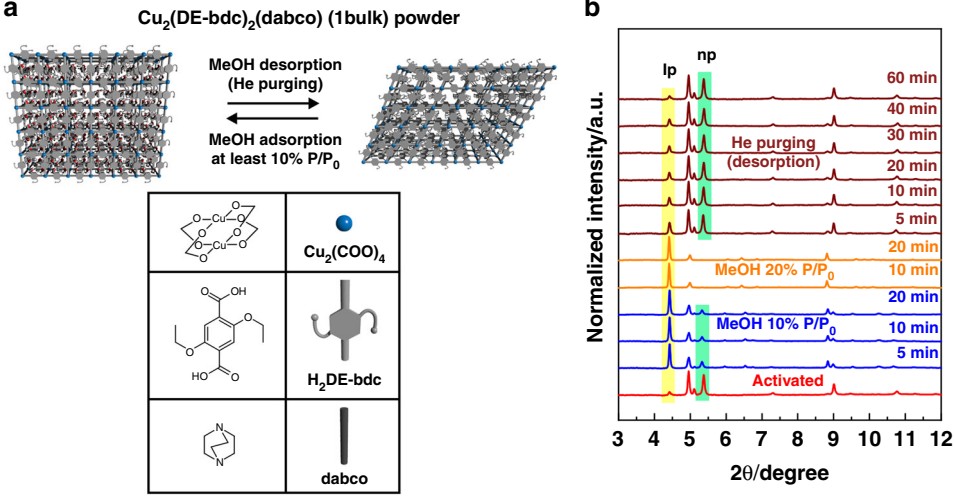

**Fig. 2** Structural responsiveness of **1bulk** upon methanol sorption. **a** Schematic representation of the breathing transformation of **1bulk** upon methanol sorption and the representations of its structural components; **b** In-situ monitoring of GIXRD patterns (X-ray wavelength of 0.827 Å) during methanol sorption at 25 °C. The structural change from the activated, **np** form to the solvated, **lp** form is initiated by adsorption of 10% $P/P_0$ methanol and is complete after adsorption of at least 20% $P/P_0$. Desorption of methanol by purging with dry He gas flow regains the activated, **np** form, which indicates the reversible structural responsiveness of **1bulk**

recorded during dynamic methanol adsorption at 25 °C. Herein, the structural response of **1tf$_x$** are studied during 4 different stages of the methanol sorption process: (1) the as-synthesised, solvated stage, (2) the activation procedure by purging with dry He gas (transformation from solvated to guest-free (activated) state), (3) the methanol adsorption at various $P/P_0$ (incremental transformation from guest-free to solvated state), and (4) the desorption process (solvated back to guest-free state). The responsiveness of

**1tf$_x$** crystallites is investigated using out-of-plane and in-plane GIXRD profiles.

The out-of-plane GIXRD profiles of **1tf$_x$** (on —COOH functionalised QCM substrates, Fig. 4) show a significantly different structural responsiveness upon methanol sorption compared to **1bulk**. Moreover, we observe a dependency of the structural flexibility on the total number of LPE fabrication cycles. The crystallites in **1tf$_{40}$** remain in the **lp** form regardless of the

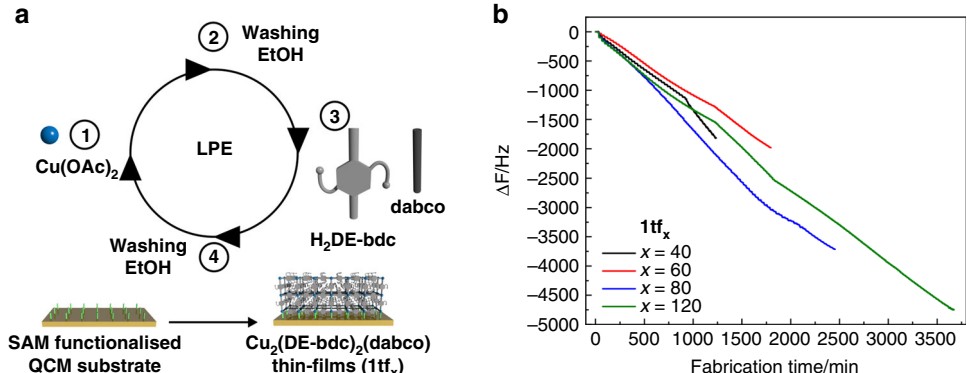

**Fig. 3** LPE growth of **1tf$_x$** and changes in the QCM oscillator frequency (*F*) as a function of time during the growth. **a 1tf$_x$** were fabricated via a stepwise LPE process in the continuous flow mode at 40 °C for 40 (black plot), 60 (red plot), 80 (blue plot) and 120 cycles (green plot). In each cycle, the QCM substrate was alternatingly exposed to the solutions of metal and mixed organic precursors followed by washing with ethanol after each precursor dosing step; **b** In-situ monitoring of the QCM frequency change indicates overall linear-growth profiles with respect to the number of repeated deposition cycles. Increasing the total number of deposition cycles shows a decrease of frequency. According to the Sauerbrey Equation, this observation indicates an increase of mass (i.e., MOF) anchored to the QCM substrate (corresponding to a thicker film with larger crystallite dimension)

treatments (Fig. 4a and Fig. 5a). We attribute this to the strong chemical binding of the MOF crystallites and the SAM-functionalised surface. This strong surface interaction within **1tf$_{40}$** presumably inhibits the **lp**-to-**np** transformation, either because the strain caused by the structural transformation at the interface cannot be compensated by the beneficial side chain-side chain interactions of the **np** form or it is a size effect as observed in the work of Kitagawa and co-workers on Cu$_2$(bdc)$_2$(bipy)[43].

Contrary, **1tf$_{60}$**, **1tf$_{80}$** and **1tf$_{120}$** exhibit a structural transformation upon methanol sorption demonstrated by a marked change of the GIXRD patterns. Specifically, the films change from the solvated, **lp** form to a mixture of both **lp** and **np** forms after activation by purging the samples with He gas (red plots, Figs. 4b–d). Upon activation, on one hand, the crystallite domains anchored closely to the substrate surface still remain in the guest-free, **lp** form due to the restricted framework flexibility as observed for **1tf$_{40}$**. On the other hand, the crystallite domains at a higher distance to the surface, which have less interaction to the substrate, can undergo a breathing transformation and then change from the solvated, **lp** form to the guest-free, **np** form (Fig. 5b). Notably, adsorption of 10% *P/P$_0$* methanol into the larger-dimension **1tf** films cannot initiate the framework transition towards the solvated, **lp** form as observed for **1bulk** (blue plots, Figs. 4b–d). Methanol with a *P/P$_0$* of at least 20% (orange, green and purple plots, Figs. 4b–d) is required to trigger the structural flexibility, highlighting the unique effect of anchoring flexible MOFs onto a surface (Fig. 5b). These observations emphasise that the observed **lp** and **np** forms in activated **1tf$_{60}$**, **1tf$_{80}$** and **1tf$_{120}$** are not a mixture of two different forms of the activated, **np 1bulk** deposited on top of the activated **lp 1tf** (see Supplementary Fig. 13). The framework flexibility in **1tf** is reversible by purging with He, showing the transformation back to the activated stage consisting of both **lp** and **np** forms.

Top-view scanning electron microscopy (SEM) images (Supplementary Figs. 15 and 16) show the cubic MOF crystal particles within the **1tf$_x$** samples (in the lateral dimensions in parallel to the substrate). Analysis of crystal particle size distribution (Supplementary Fig. 17) indicates that **1tf$_{40}$** has significantly smaller particle size (80% of particle size distribution ranging within 100–300 nm with an average size of about 200 nm) than **1tf$_{60}$**, **1tf$_{80}$** and **1tf$_{120}$** (80% of particle size distribution ranging within 250–650 nm with an average size of about 400 nm). Moreover, cross-sectional SEM images (Supplementary Fig. 18) also reveal a significant difference of the film thickness of **1tf$_{40}$** (430 nm) from

**1tf$_{60}$** (740 nm), **1tf$_{80}$** (840 nm) and **1tf$_{120}$** (1000 nm). On the same area of QCM sensors, anchoring of the smaller crystal particles on the substrate (**1tf$_{40}$**) may show a significant influence of the surface interaction over the whole particles. This consequence leads to a higher activation barrier for the framework breathing in **1tf$_{40}$** and hence all the crystallite domains remain in the **lp** form even in the activated, guest-free condition. Unlike **1tf$_{40}$**, the crystallite domains which are less affected by the surface interaction within the thicker **1tf$_{60}$**, **1tf$_{80}$** and **1tf$_{120}$** films consisting of larger particles can undergo the guest-induced framework transition.

Additionally, methanol adsorption isotherms measured at 25 °C on an environmentally controlled QCM indicate different adsorption profiles within the **1tf$_x$** series (Fig. 6). A marked difference of the isotherm shape can clearly be identified. Specifically, **1tf$_{40}$** exhibits a single-step methanol adsorption signalling the presence of only the **lp** form after activation due to a restriction of the framework flexibility. In contrast, **1tf$_{60}$**, **1tf$_{80}$** and **1tf$_{120}$** show two-step adsorption profiles, which are typically indicative of a guest-induced phase transition. The first step (*P/P$_0$* from 0 to 0.15) is attributed to the adsorption of methanol into both the existing **lp** and **np** forms of the activated samples, without inducing the framework transformation. A rapid increase of the adsorbed methanol amount at *P/P$_0$* of 0.15 indicates the threshold pressure that initiates the **np**-to-**lp** frameworks transition, leading to a further increase of the amount adsorbed. At high *P/P$_0$*, all the **1tf$_x$** samples show nearly-identical adsorption values (of approximately 7.5 mmol of methanol per gram of MOF), since all the crystallites are solely presented in the **lp** form. This observation also reveals the similar accessibility of the pores of the fabricated **1tf$_x$** samples, which is independent of the thickness (crystallite dimension) of the thin-films. Note that, all the **1tf$_x$** samples rather show a single-step desorption profile (Supplementary Fig. 19).

For additional investigations, we fabricated **1tf** on pyridyl-terminated SAM-functionalised QCM substrates by employing LPE deposition for 60 cycles (named as **1tf$_{60\text{-Py}}$**). **1tf$_{60\text{-Py}}$** exhibits preferred growth along the (001)-plane of the framework, along the dabco ligand[37]. Unlike **1tf$_{60}$**, GIXRD profiles during methanol sorption of **1tf$_{60\text{-Py}}$** (Supplementary Fig. 20) present only the **lp** form in all sorption treatments. Moreover, **1tf$_{60\text{-Py}}$** exhibits a single-step methanol adsorption isotherm (Supplementary Fig. 21), with a small deviation from the typical type-I adsorption isotherm at *P/P$_0$* of 0.15. This implies the possibility of partial

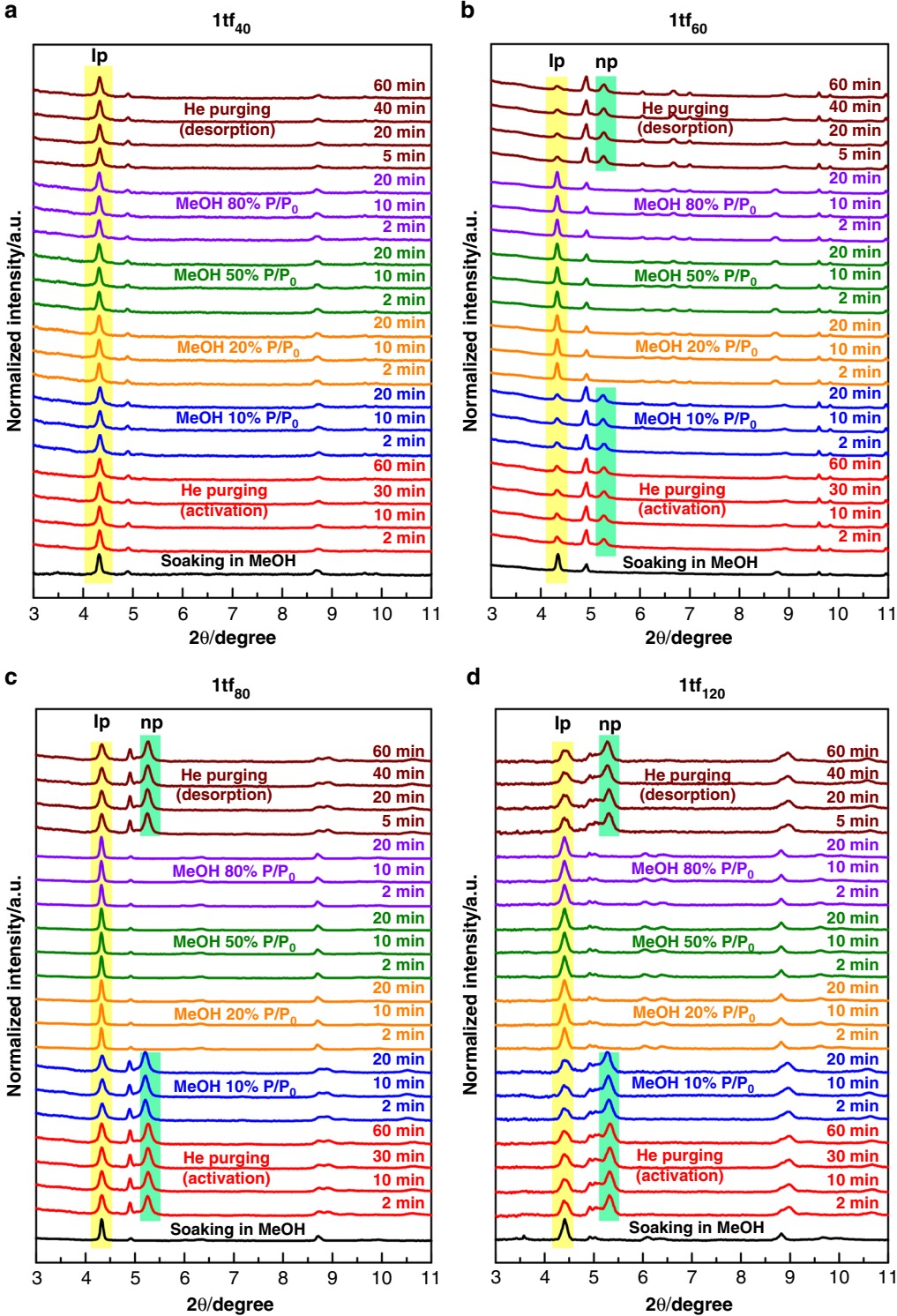

**Fig. 4** Out-of-plane GIXRD profiles of the **1tf$_x$** materials during methanol sorption at 25 °C. Herein, **a 1tf$_{40}$**, **b 1tf$_{60}$**, **c 1tf$_{80}$** and **d 1tf$_{120}$** were fabricated on — COOH functionalised QCM substrates, which show a preferred growth along the (110) plane[37]. Yellow and green highlighted bars represent the most intense diffraction peaks belonging to the **lp** and **np** phase, respectively. Herein, **1tf$_x$** exhibit a significantly different structural flexibility upon methanol sorption than **1bulk**. Moreover, a marked dependence of the structural flexibility on the total number of fabrication cycles is observed (see also Fig. 5)

flexibility, however there is no phase transition observed in the GIXRD. It strongly points towards that the breathing is more restricted in **1tf$_{60-Py}$** than in **1tf$_{60}$**. In **1tf$_{60-Py}$**, the whole Cu$_2$(DE-bdc)$_2$ grids are fixed parallel to the surface. The wine rack-like movement during the breathing transition would considerably affect the anchoring, disturbing the orientation of the SAM or putting a lot of stress on the interface. Hence we believe that the energy penalty caused at the interface cannot be compensated by the beneficial side chain-side chain and side chain-metal centre interactions of the **np** state. In other words, the flexibility is

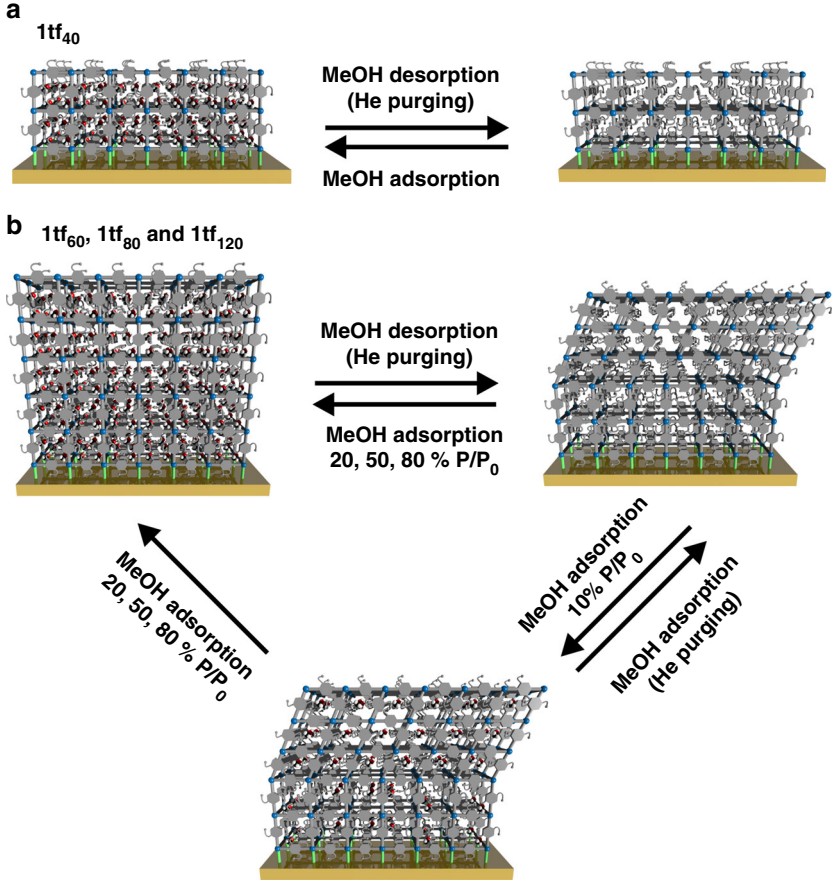

**Fig. 5** Schematic representation of the structural response of **1tf$_x$** upon methanol sorption. **1tf$_x$** materials exhibit a unique structural transition depending on the total number of LPE fabrication cycles (i.e., the crystallite dimension and/or film thickness). **a 1tf$_{40}$** remains in the **lp** form, regardless of the treatments, indicating the restriction of the framework transition when anchored on the surface. **b** The films with larger crystallite-dimension (**1tf$_{60}$**, **1tf$_{80}$** and **1tf$_{120}$**) surprisingly reveal a unique structural flexibility. The crystallites domains close to the interface with the substrate surface remain in the **lp** form, while the domains in sufficient distance to the surface can undergo a **lp**-to-**np** breathing transition upon activation. Hence, both the **lp** and **np** forms are presented at the activated stage. Adsorption of methanol vapour of at least 20% $P/P_0$ is required for the reverse **np**-to-**lp** structural transition to the solvated, **lp** stage. Unlike the case of **1bulk**, 10% $P/P_0$ of methanol vapour can be adsorbed in **1tf$_x$** but cannot induce the structural transition

inhibited if the growth direction of **1tf** is changed. Notably, the guest-induced structural flexibility of **1tf** depends not only on the crystallite dimension but also on the orientation of the crystallites anchored on the surface.

Furthermore, LPE is used for fabrication of **2tf$_{60}$**, **2tf$_{80}$** and **2tf$_{120}$** on the —COOH functionalised QCM substrates. According to in-situ GIXRD profiles (Supplementary Figs. 23 and 24), the **2tf$_x$** materials also exhibit structural responsiveness upon methanol sorption in a similar manner to the **1tf$_x$** materials, which clearly indicates the guest-induced framework transition in related alkylether-functionalised MOF thin-films. In contrast, the pristine Cu$_2$(bdc)$_2$(dabco) (**3**) thin film fabricated by LPE for 60 cycles (**3tf$_{60}$**) does not show any structural reorganisation upon methanol sorption (Supplementary Fig. 25). This further emphasises that the alkylether pendent side chains are necessary for inducing structural flexibility upon polar-guest sorption into the Cu-based layered-pillared MOFs, as well as their thin-films.

## Discussion

We employed for the first time the stepwise LPE process to anchor a nanolayer of flexible layered-pillared MOF crystallites onto SAM-functionalised QCM substrates. Variation of the SAM-functionalised surface and the total number of fabrication cycles controls the

crystallite orientation and dimension (and/or thickness) of the thin-films, as shown in previous works. Remarkably, the crystallite dimensions have a distinct effect on the structural dynamics. While the flexibility is inhibited for very thin films (40 deposition cycles), thicker films (60–120 deposition cycles) show a **lp**-to-**np** phase transition upon polar-guest desorption which is reversible upon re-adsorption. Additionally, the orientation of the film has a crucial impact on the transition, while samples grown along the (110) lattice plane show structural dynamics, these are inhibited for MOFs grown along the (001) lattice plane. We attribute the restriction of structural flexibility to the strong interaction between the crystallites and the SAM-functionalised substrate leading to a higher activation barrier for initiating the flexibility (such as in the low crystallite-dimension **1tf$_{40}$** and the (001)-oriented **1tf$_{60-Py}$**). The breathing transformation occurs in the crystallite domains further away from the substrate interface, as the influence of the interface gets less pronounced. However, the effect of anchoring the flexible MOF crystallites onto the surface leads to increased threshold vapour pressures of methanol for initiating the **np**-to-**lp** transformation with respect to the bulk samples. In all, we provide a toolkit for the targeted preparation of MOF thin films with adjustable framework flexibility and demonstrate the fundamental understanding of breathing transitions within MOF thin films. The findings at hand are of relevant interest for specific applications such as MOF-thin film based sensors. A further extension towards well-controlled

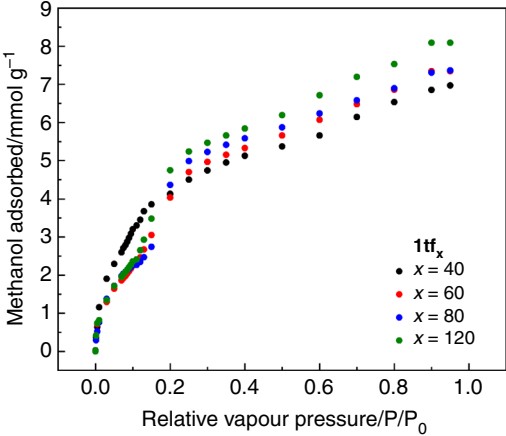

**Fig. 6** Methanol adsorption isotherms of **1tf_x** measured at 25 °C on an environmentally controlled QCM. **1tf₄₀** (black plots) exhibits a single-step methanol adsorption isotherm, whereas **1tf₆₀** (red plots), **1tf₈₀** (blue plots) and **1tf₁₂₀** (green plots) show two-step adsorption. The step in the adsorption profiles starting at $P/P_0 = 0.15$ indicates the threshold pressure that initiates the methanol-dependent **np**-to-**lp** structural flexibility. Herein, it reveals the relation between the guest-induced structural transitions of **1tf_x** on the dimension of the MOF crystallites anchored to the surface (which is proportional to the number of LPE fabrication cycles, as well as the film thickness)

crystallite orientation[54–56] and a combination of various responsivities by means of heterostructured architectures[57,58] is currently under development for the preparation of multiple-stimuli-responsive MOF-based devices.

## Methods

**MOF thin-film fabrication**. Au-coated QCM sensors (Q-Sense, Au electrode, diameter 14 mm, thickness 0.3 mm and fundamental frequency ca. 4.95 MHz) were used as substrates. Prior to the thin-film fabrication, the substrates were immersed in 20 µM solution of 16-mercaptohexadecanoic acid (MHDA) in ethanol mixed with 5% v/v acetic acid for 24 h to generate a —COOH terminated surface. In order to generate a —pyridyl terminated surface, a 20 µM solution of 4-(4-pyridyl)phenylmethylthiol (PMBT) in ethanol was used instead of the MHDA solution. **1tf_x** and **2tf_x** ($x = 40$, 60, 80 and 120 cycles) were fabricated by the LPE process at controlled temperatures of 40 °C using an automated QCM instrument (Q-Sense E4 Auto) operated in the continuous flow mode with a constant flow rate of 100 µL min⁻¹. In each deposition cycle, the —COOH terminated QCM substrate was alternately exposed to the precursor solutions as follow: Cu(OAc)₂·H₂O (0.5 mM in ethanol) 10 min, ethanol 5 min, the mixed organic linkers (H₂DE-bdc + dabco for **1tf_x** and H₂BME-bdc + dabco for **2tf_x**, 0.2 mM in ethanol) 10 min, and finally ethanol 5 min. During the fabrication, the QCM frequency change was monitored in-situ. In addition, **1tf** was also prepared on the —pyridyl terminated QCM by LPE for 60 cycles (**1tf₆₀-Py**) in order to study the influence of crystallite orientation on the structural responsiveness of the MOF films. For comparison, the parent Cu₂(bdc)₂(dabco) thin film was also prepared by LPE for 60 cycles (**3tf₆₀**).

**In-situ synchrotron X-ray diffraction during methanol adsorption**. Crystalline phase and structural flexibility during methanol sorption of the MOF powders and thin-films were identified by GIXRD (Beamline BL9 of the DELTA synchrotron radiation source, Dortmund, Germany. An X-Ray energy of 15.0 keV corresponding to a wavelength of 0.827 Å was used. The angle of incidence was 0.6°)[59]. In order to investigate the structural flexibility induced by polar-guest (i.e., methanol) sorption at ambient temperature (25 °C), the sample stage was equipped with a heatable graphite dome-type unit (DHS 1100, Anton Paar). This sample stage was further connected with the custom-built He gas flow system, which could semi-quantitatively control the dosing amount of methanol vapour into the sample chamber. The MOF sample was placed in the sample chamber and covered with the graphite dome. The temperature of the sample chamber was controlled at 25 °C during the whole experiment. In-situ GIXRD patterns were recorded in different stages of the methanol sorption process as follows: (1) the as-synthesised, solvated stage (mimicking by drop methanol to the sample); (2) the activation (by purging the sample with He gas); (3) the methanol adsorption with 10, 20, 50 and 80% $P/P_0$; and (4) the desorption (by purging with He gas flow). To control the $P/P_0$, the pressure-controlled valves at the pure He gas line and the one that passed through the methanol vaporiser were adjusted to change the mixing ratio, while keeping the total pressure at the end point behind the sample chamber constant (the schematic setup is shown in Supplementary Fig. 4).

**Characterisation of MOF thin-films**. Surface morphology and surface coverage were investigated by a field emission scanning electron microscopy (FE-SEM, ZEISS Gemini Sigma 300 VP). Cross-sectional SEM images were measured by a JEOL JSM-7500F operated in Gentle Beam mode without sputtering in order to determine the film thickness. Sorption properties of the MOF films were carried out on an environmental-controlled QCM (BEL-QCM-4 instrument, MicrotracBEL Corp.) at controlled temperature of 25 °C using methanol as probe molecules. Prior to the sorption measurements, the films were activated in-situ within the BEL-QCM instrument by purging with a dry He gas with a flow rate of 100 sccm for 2 h until the change of QCM frequency was stable within the range of ±5 Hz in 20 min. Then the mass of the MOF film was recorded by a conversion of the difference between the QCM frequency at the final activation and the fundamental frequency of the SAM-functionalised QCM substrates according to Sauerbrey's equation. After that, methanol sorption isotherms at 25 °C were collected by varying $P/P_0$ of saturated methanol vapour in He gas flow from 0.0 to 95.0%.

## Data availability

The authors declare that the data supporting the findings of this study are available within the paper and its supplementary information file, which is available in the online version of the paper. Correspondence and requests should be addressed to R.A.F.

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

## Acknowledgements

This work was initially supported by the Cluster of Excellence RESOLV (EXC 1069) and further by DFG FOR 2433 FLEXMOF funded by the German Research Foundation (DFG). DELTA is acknowledged for providing synchrotron radiation at beamline BL9 for grazing incidence X-ray diffraction measurements. S.W. acknowledges the Royal Thai Government under the Ministry of Science and Technology for a Ph.D. scholarship, the International Realisation Budget (IRB) of the Research School Plus, Ruhr-University Bochum (GSC 98) for international travel and research funding, and the Technical University of Munich for a Postdoctoral Fellowship. A.S. acknowledges the German Research Foundation for a Postdoctoral Fellowship. Katja Rodewald, Wacker-Chair of Macromolecular Chemistry, Technical University of Munich is acknowledged for measuring the cross-sectional SEM of the thin-film samples.

## Author contributions

S.W. conceived, designed and performed the project under supervision of R.A.F; A.S. synthesised the organic linkers and bulk powder samples; S.W., A.S., I.H., M.T., K.E. performed the in-situ GIXRD at DELTA synchrotron facility; S.W. and A.L.S. collected the adsorption data and SEM images; C.S. and M.P. helped designing the setup for in-situ GIXRD during adsorption and provided helpful discussion and support during the measurement at DELTA; A.S. and S.J.B. performed the Pawley refinements of GIXRD patterns of the powder samples; S.W., A.S., G.K. and R.A.F. co-wrote the paper.

## Additional information

**Competing interests:** The authors declare no competing interests.

