## [Peer Review File · Nature Communications]

Reviewers' comments:

Reviewer #1 (Remarks to the Author):

Fischer et al. report crystal-size dependent flexibility upon methanol adsorption in Cu-based metal-organic framework (MOF) thin film. This manuscript itself has been clearly written and the results presented is interesting for broad readership of Nature Communications, however, this manuscript lacks basic characterization of bulk MOF and detailed discussion on the crystal-size effect of liquid-phase epitaxy (LPE) as shown below.

First of all, the authors describe the thin film of Cu-MOF based on DE-bdc (1) and BME-bdc (2) ligands. The as-synthesized bulk 1 and 2 are isostructural, which can be easily seen in Supplementary Figure 5 (PXRD patterns). And the structural characterization of as-synthesized form was already done by single-crystal X-ray diffraction for Zn-based analogue as discussed in Refs 33 and 35. However, the structural characterization of their activated forms were completely missing. I am wondering where the structural evidence of Figure 1 is? I think this point is very unclear for readers and critical for publication. In Ref. 35, the authors tentatively identified the activated forms of Cu, Ni, and Co-based analogues by using Pawley fitting. From the result of that paper, it was found that the Cu compound (2) is crystallized in different space group from that of Zn compound. Also, the PXRD patterns of activated 1 and 2 are different from each other as shown in Supplementary Figure 5. As you know, the Pawley fitting is not enough for direct structural characterization because the Pawley fitting is not the refinement on structural parameters (i.e., atomic coordinates, isotropic thermal displacement parameters, and site occupancies) but the refinement on the lattice constant and profile function. To show the structural evidence for activated forms of 1 and 2, the authors are strongly requested to perform Rietveld refinements (or single-crystal XRD), and should add the fitting results and cif files including refined fitting parameters for revised supplementary information. I think the schematic of this paper shown in Figure 1 seems to lack the experimental evidence without detailed structural analysis on activated form at this stage.

Second, the authors should add the detailed discussion on LPE-cycle dependency of crystal size of thin films. In this paper, the authors discuss the LPE-cycle dependency of crystal size using SEM (Supplementary Figures 12 and 13) images for thin films of 1, however, from out-of-plane and in-plane GIXRD patterns (Figure 4 and Supplementary Figure 11) of thin films of 1, I could not see the successive increase of crystal size (usually calculated from FWHM of diffraction peak) upon increase of LPE cycle. How do the authors explain the effect of LPE-cycle dependency on crystal size in 1? Also, the authors should add the data of crystal size estimated from XRD results on each thin film, and compare the LPE-cycle dependent crystal size with the SEM results. In addition, the authors are recommended to add the data set of SEM images and in-plane GIXRD on thin films of 2 same as thin films of 1.

I think major revision is needed.

Reviewer #2 (Remarks to the Author):

The authors present, in general, a very easy to read and delightful article on the stepwise LPE process used to anchor a layer of a flexible pillared MOF. The authors have gone to great lengths to understand how functionalising the framework, the growth direction and number of fabrication cycles effects the structural dynamics. In particular, the in-situ synchrotron grazing incidence X-ray diffraction (GIXRD) developed to monitor guest adsorption by using a custom-built, semi-quantitative vapour adsorption unit (as seen in Supplementary Figure 4), was very very neat, and could be utilised by a number of groups outside of the field of MOF-chemistry. The in plane and out of plane GIXRD measurements to understand how the MOF is orientated on the surface is also rarely done, in general, never mind for MOFs and should be applauded. The difference seen on the uptake of MeOH and the transition from the lp to np phases on 1tfx compared to the bulk was also very well described.

I only have a few minor comments;

Page 2; the authors discuss 'flexible' crystals. It may be worth mentioning the recent publication by Clegg et al, (Nature Chemistry volume10, pages65–69 (2018))

Page 4, the use of the word 'comprehension' is just odd, and used incorrectly.

Page 4, remove 'the' in the sentence starting 'Moreover, the LPE allows' and reword the sentence.

Page 4, state the exact size, and reword the section that states 'to the nanometres-size'

Figure 3a and supplementary figure 2b are identical, are both needed?

The authors say that '1t60-Py exhibits a strongly preferred growth orientation along the (001) plane', how was this determined?

Page 12, 'fixed parallelly' needs changed to 'fixed parallel'

Overall, I think this paper, would attract a wide audience and deserves publication.

.....

Reviewer 1:

Fischer et al. report crystal-size dependent flexibility upon methanol adsorption in Cu-based metal-organic framework (MOF) thin film. This manuscript itself has been clearly written and the results presented is interesting for broad readership of Nature

Communications, however, this manuscript lacks basic characterization of bulk MOF and detailed discussion on the crystal-size effect of liquid-phase epitaxy (LPE) as shown below.

***Answer:** We thank the Reviewer for the comments and the interest in our work. We are particularly grateful for pointing out the unclarities regarding to the structural analysis which we believe to have fully addressed in our response, see below.*

.....
First of all, the authors describe the thin film of Cu-MOF based on DE-bdc (1) and BME-bdc (2) ligands. The as-synthesized bulk 1 and 2 are isostructural, which can be easily seen in Supplementary Figure 5 (PXRD patterns). And the structural characterization of as-synthesized form was already done by single-crystal X-ray diffraction for Zn-based analogue as discussed in Refs 33 and 35. However, the structural characterization of their activated forms were completely missing. I am wondering where the structural evidence of Figure 1 is? I think this point is very unclear for readers and critical for publication.

***Answer:** We thank the reviewer for raising this important point. The schematic shown presents structures from established findings from the literature, based on Pawley profile fits, theory, solid state NMR and lattice parameters which have shown and characterized the wine-rack type flexibility in these MOFs (in the bulk) several times. Thus, in our work we can build upon this previous well established data.*

*In more detail, referring to references 33 and 35 in the previous-submitted manuscript (and similarly references 37 and 39 in the revised manuscript, single crystal XRD was used to identify the structural evidence of the as-synthesized (solvated) form of Zn-based alkoxyether-functionalized layered-pillared MOFs and the Pawley refinements were used to identify the change of lattice parameters with respect to the breathing phenomena upon guest adsorption and desorption process. This refinement indicated the induced wine-rack-like motion which changes the square $M_2(\text{fu-bdc})_2$ grids in the solvated, **lp** form into the rhombic grids in the activated, **np** form, while the dabco-containing axis remains largely unchanged, resulting in a change of the XRD patterns.*

*Herein, we performed the detailed Pawley refinements of the lattice parameters of the MeOH-solvated and activated forms of **1bulk** and **2bulk** materials from their GIXRD patterns (using synchrotron X-ray wavelength of 0.827 Å). The patterns and fits were added to this revised manuscript as Supplementary Figure 6 and 7 and the obtained cell parameters are summarized in Supplementary Table 1.*

*In good agreement with references 33 and 35, we observe from the Pawley refinements that **1bulk** (as well as **2bulk**) reveals an increase of lattice parameter a and shrinkage of lattice parameter b , while lattice parameter c remains mostly constant upon MeOH*

*desolvation. Consequently, these changes of the unit cell parameters lead to a reduction of the total volume of the unit cell, indicating the breathing transition upon guest sorption (from the solvated, **lp** form to the activated, **np** form) in a similar manner as reported previously in the literatures (Ref. 37 and Ref. 39 in this revised manuscript).*

*These evidences are used to confirm the structural transition so-called guest-induced breathing of the materials **1bulk** and **2bulk** as well as **their thin-films (SURMOFs)** during the sorption of methanol.*

This more extended comparison and discussion of data was added to the revised manuscript on Page 7.

.....
In Ref. 35, the authors tentatively identified the activated forms of Cu, Ni, and Co-based analogues by using Pawley fitting. From the result of that paper, it was found that the Cu compound (2) is crystallized in different space group from that of Zn compound. Also, the PXRD patterns of activated 1 and 2 are different from each other as shown in Supplementary Figure 5.

As you know, the Pawley fitting is not enough for direct structural characterization because the Pawley fitting is not the refinement on structural parameters (i.e., atomic coordinates, isotropic thermal displacement parameters, and site occupancies) but the refinement on the lattice constant and profile function. To show the structural evidence for activated forms of 1 and 2, the authors are strongly requested to perform Rietveld refinements (or single-crystal XRD), and should add the fitting results and cif files including refined fitting parameters for revised supplementary information. I think the schematic of this paper shown in Figure 1 seems to lack the experimental evidence without detailed structural analysis on activated form at this stage.

***Answer:** We do agree that Rietveld refinement or measuring a single crystal XRD provides more information in order to depict the actual structure of the activated form of **1bulk** and **2bulk**. However, the structural change of these functionalized layered-pillared MOF systems have been proven several times in the literatures based on the evidences from the Pawley refinements, the solid state NMR and the supporting theoretical calculation (Ref. 37-40). Hence, we followed the similar way to prove the evidence of framework flexibility of the obtained MOFs in both powder and thin-film forms. The detailed identification of atomic coordinates within the structures may help for better understanding of the breathing phenomena in general. However, it will not significantly contribute to the characterization of guest-induced structural flexibility in case of surface mounted thin films or crystallites of such kind of MOFs, which is the topic of our study.*

*In addition, obtaining good quality single crystals of the activated **1bulk** and **2bulk** (or even the analogous Cu-based layered-pillared MOFs) still remains a great challenge. We are not aware of a SCXRD of any $\text{Cu}_2(\text{fu-bdc})_2(\text{dabco})$ derivatives since these*

materials tend to nucleate too fast (which is advantageous for the thin film growth by liquid-phase epitaxial growth). Obtaining single crystals of activated $\text{Cu}_2(\text{fu-bdc})_2(\text{dabco})$ derivative is extremely challenging. Note, we do have a single crystal structure of a closely related and activated system, $\text{Zn}_2(\text{fu-bdc})_2(\text{dabco})$ (fu-bdc = 2,5-dipentoxo-1,4-benzene dicarboxylate), see Ref. 40. However, this was a lucky case for SCXRD and this particular compound is not breathing due to the large pentoxo substituents filling up the pore space. The preparation of high quality films of breathing fu-bdc MOFs is already very challenging and being able to also obtain SCXRD from the bulk phase in the activated form is difficult, as the properties that induce smooth film growth (fast nucleation kinetics) are contrary to the ones that allow for single crystal growth (slow nucleation kinetics).

Moreover, there is a limitation to obtain a perfect intensity distribution in the GIXRD setup leads to a limitation to identify the structural parameters of the activated form of **1bulk** and **2bulk** by Rietveld refinement (as suggested) properly. It is also not at all trivial to solve these structures from powder XRD patterns due to the possibility to have crystal cracks, high amount of disorders in the polycrystalline powder samples, which definitely affected the powder XRD patterns.

.....

Second, the authors should add the detailed discussion on LPE-cycle dependency of crystal size of thin films. In this paper, the authors discuss the LPE-cycle dependency of crystal size using SEM (Supplementary Figures 12 and 13) images for thin films of 1, however, from out-of-plane and in-plane GIXRD patterns (Figure 4 and Supplementary Figure 11) of thin films of 1, I could not see the successive increase of crystal size (usually calculated from FWHM of diffraction peak) upon increase of LPE cycle. How do the authors explain the effect of LPE-cycle dependency on crystal size in 1? Also, the authors should add the data of crystal size estimated from XRD results on each thin film, and compare the LPE-cycle dependent crystal size with the SEM results. In addition, the authors are recommended to add the data set of SEM images and in-plane GIXRD on thin films of 2 same as thin films of 1. I think major revision is needed.

Answer: We would like to emphasize that we mentioned the different crystallite dimensions of the **1tf_x** samples prepared by varying the number of LPE deposition cycles. This crystalline dimension is more related to the crystal particle size (top-view SEM images) and the film thickness (cross-sectional SEM images) but not the calculated crystallite size based on XRD patterns using the Scherrer equation.

In detail, since the observed particle size within the **1tf_x** samples are in the sub-micrometer range, a calculation of the crystallite size based on the XRD data using Scherrer equation is not properly applicable. Moreover, the peak broadening of the GIXRD patterns of **1tf_x** samples may be affected from the experimental set-up, leading to the non-significant difference of the calculated crystallite size (Supplementary Figures

12). Hence, the calculated crystallite size from Scherrer equation is not used for the discussion of the influence on the structural flexibility reported herein. Note that, this discussion was added to the revised manuscript on Page 9.

In this revision, we show the particle size distribution (measured from top-view SEM images) and the film thickness (cross-sectional SEM micrographs) in Supplementary Figures 15-18 and summarized all the obtained sizes in Supplementary Table 2. These observations clearly indicate that the **1tf₄₀** has significantly smaller particle sizes (in both lateral dimensions and thickness) from **1tf₆₀**, **1tf₈₀** and **1tf₁₂₀**. This evidence reveals that anchoring of the smaller crystal particles on the substrate (**1tf₄₀**) shows a significant influence of the surface interaction on the whole particles, leading to a restriction for the framework flexibility. Unlike **1tf₄₀**, the crystallite domains which are less affected by the surface interaction within the thicker **1tf₆₀**, **1tf₈₀** and **1tf₁₂₀** films can undergo the guest-induced framework flexibility. Note that, this discussion was added to the revised manuscript on Page 11-12.

Reviewer 2:

The authors present, in general, a very easy to read and delightful article on the stepwise LPE process used to anchor a layer of a flexible pillared MOF. The authors have gone to great lengths to understand how functionalising the framework, the growth direction and number of fabrication cycles effects the structural dynamics. In particular, the in-situ synchrotron grazing incidence X-ray diffraction (GIXRD) developed to monitor guest adsorption by using a custom-built, semi-quantitative vapour adsorption unit (as seen in Supplementary Figure 4), was very very neat, and could be utilised by a number of groups outside of the field of MOF-chemistry. The in plane and out of plane GIXRD measurements to understand how the MOF is orientated on the surface is also rarely done, in general, never mind for MOFs and should be applauded. The difference seen on the uptake of MeOH and the transition from the lp to np phases on 1tfx compared to the bulk was also very well described.

Answer: *We are very grateful for receiving this positive evaluation from the reviewer.*

I only have a few minor comments;

Page 2; the authors discuss 'flexible' crystals. It may be worth mentioning the recent publication by Clegg et al, (Nature Chemistry volume10, pages65–69 (2018))

Answer: *The suggested one as well as some other relevant references have been added to this revised manuscript (highlighting the added references in yellow in the revised manuscript).*

Page 4, the use of the word 'comprehension' is just odd, and used incorrectly. Page 4, remove 'the' in the sentence starting 'Moreover, the LPE allows' and reword the sentence.

Answer: *The sentences have been modified as follow;*

"The implementation of MOFs as thin films has received increased attention over the last few years,⁴⁶⁻⁴⁷ since it is imperative for the advancement of MOF-based devices."

"Moreover, the crystallite orientation can be controlled within LPE process by varying the surface functionality (e.g. by using self-assembled monolayer (SAM) of organothiols featuring different terminal-groups).⁴⁹"

Page 4, state the exact size, and reword the section that states 'to the nanometres-size'

Answer: *The exact size has been added to the sentence as shown below;*

*"Remarkably, downsizing of the non-porous $Fe(py)_2[Pt(CN)_4]$ ($py = pyridine$) to a thin film of 16 nanometers thickness initiates a gate-opening structural transformation (**cp-to-op**) induced by the lower potential energy barrier.⁵³"*

Figure 3a and supplementary figure 2b are identical, are both needed?

Answer: *We would like to keep the Supplementary Figure 2 (A and B) for better clarification of the whole LPE process for the general readers who may not be familiar with the method.*

The authors say that '1tf60-Py exhibits a strongly preferred growth orientation along the (001) plane', how was this determined?

Answer: *We fabricated **1tf** on pyridyl-terminated SAM-functionalized QCM substrates by employing LPE deposition for 60 cycles (named as **1tf_{60-Py}**). **1tf_{60-Py}** exhibits a preferred growth along the dabco-related orientation of the framework, which can be assigned to the (001)-orientation by referring to the previously-reported Zn-based analogous structure³⁷ as well as referring to the Pawley refinements of the **1bulk** sample (Supplementary Figure 6), which have been done in the similar way as mentioned in the literature.³⁷*

Page 12, 'fixed parallely' needs changed to 'fixed parallel'

Answer: *The words have been corrected.*

Overall, I think this paper, would attract a wide audience and deserves publication.

Answer: *Thank you very much.*

REVIEWERS' COMMENTS:

Reviewer #1 (Remarks to the Author):

In this paper, Fischer et al. report crystal-size dependent flexibility upon methanol adsorption in Cu-based metal-organic framework (MOF) thin film. I have just read their revised manuscript. I think the authors made an effort to respond all the referees' comments and our concerns have been addressed properly. I still think Rietveld analysis for bulk state (not thin film state) would offer valuable information about flexibility of present material. But, this version of manuscript is acceptable for publication in Nature Communications.

Response to Reviewers' Comments

REVIEWERS' COMMENTS:

Reviewer #1 (Remarks to the Author):

In this paper, Fischer et al. report crystal-size dependent flexibility upon methanol adsorption in Cu-based metal-organic framework (MOF) thin film. I have just read their revised manuscript. I think the authors made an effort to respond all the referees' comments and our concerns have been addressed properly. I still think Rietveld analysis for bulk state (not thin film state) would offer valuable information about flexibility of present material. But, this version of manuscript is acceptable for publication in Nature Communications.

Answer: We would like to acknowledge the valuable suggestion from the reviewers which are very useful for us to revise our manuscript. We do agree that the Rietveld analysis of PXRD patterns of the bulk samples would give details on the structure of the bulk materials. Thank you very much.

.....